# Intra-Abdominal Fat Adipocyte Hypertrophy through a Progressive Alteration of Lipolysis and Lipogenesis in Metabolic Syndrome Rats

**DOI:** 10.3390/nu11071529

**Published:** 2019-07-05

**Authors:** Israel Pérez-Torres, Yolanda Gutiérrez-Alvarez, Verónica Guarner-Lans, Eulises Díaz-Díaz, Linaloe Manzano Pech, Sara del Carmen Caballero-Chacón

**Affiliations:** 1Department of Pathology, Instituto Nacional de Cardiología “Ignacio Chávez”, Juan Badiano 1, Sección XVI, Tlalpan, México City 14080, Mexico; 2Department of Physiology, Instituto Nacional de Cardiología “Ignacio Chávez”, Juan Badiano 1, Sección XVI, Tlalpan, México City 14080, Mexico; 3Department of Reproductive Biology, Instituto Nacional de Ciencias Médicas y Nutrición “Salvador Zubirán”, Vasco de Quiroga 15, Sección XVI, Tlalpan, México City 14000, Mexico; 4Facultad de Medicina y Veterinaria y Zootecnia, Department of Physiology and Pharmacology UNAM, Av. Universidad 3000, Coyoacán, México City 04510, Mexico

**Keywords:** metabolic syndrome, adipocyte, hypertrophy, lipoprotein lipase, hormone-sensitive lipase, perilipin A

## Abstract

This study evaluates the progressive participation of enzymes involved in lipolysis and lipogenesis, leading to adipocyte hypertrophy in a metabolic syndrome (MS) rat model caused by chronic consumption of 30% sucrose in drinking water. A total of 70 male Wistar rats were divided into two groups: C and MS. Each of these groups were then subdivided into five groups which were sacrificed as paired groups every month from the beginning of the treatment until 5 months. The intra-abdominal fat was dissected, and the adipocytes were extracted. Lipoprotein lipase (LPL), hormone-sensitive lipase (HSL), protein kinases A (PKA), and perilipin A expressions were determined. The LPL and HSL activities were evaluated by spectrophotometry. Histological staining was performed in adipose tissue. Significant increases were observed in blood pressure, HOMA-IR, leptin, triglycerides, insulin, intra-abdominal fat, and number of fat cells per field (*p* = 0.001) and in advanced glycosylation products, adipocyte area, LPL, HSL activities and/or expression (*p* ≤ 0.01) in the MS groups progressively from the third month onward. Lipogenesis and lipolysis were increased by LPL activity and HSL activity and/or expression. This was associated with hyperinsulinemia and release of non-esterified fatty acids causing a positive feedback loop that contributes to the development of adipocyte hypertrophy.

## 1. Introduction

According to the World Health Organization, metabolic syndrome (MS) is defined as the prevalence of three or more of the following risk factors: obesity, hypertension, hypertriglyceridemia, hypercholesterolemia, decrease in high density lipoproteins, hyperinsulinemia, and insulin resistance (IR) [1]. However, in MS, obesity is the most prevalent sign observed [2]. The increase in adipose tissue in the body is a consequence of a hyper-caloric diet and/or a diet rich in saturated fat, in association to a low energy expenditure due to physical inactivity and/or the presence genetic factors [3]. Currently, obesity is considered as a public health problem throughout the world and, due to its complexity, there is no single reason that explains the obesity pandemic [4]. During the development of obesity, the positive energy balance promotes adipocyte hypertrophy associated to hypertriglyceridemia, an increase in very low density lipoprotein and non-esterified fatty acids (NEFAs), ectopic storage of triglycerides (TG), and IR [2].

Adipose tissue is composed of fat cells adapted to store and release NEFAs. Two variants are distinguished; the white and the brown adipose tissue [5]. Brown adipose tissue produces heat while white adipose tissue stores energy. White adipose tissue is composed of mature adipocytes and a stromal vascular fraction. It has two metabolic functions: lipogenesis and lipolysis. Lipogenesis in adipocytes is the process by which fatty acids derived from lipoproteins are esterified with glycerol to form TG which are then stored in lipid droplets. Lipolysis is the catabolism of the previously stored TG in adipocytes [6]. The key enzymes in these processes are lipoprotein lipase (LPL) and hormone sensitive lipase (HSL) respectively [7]. Decreases in HSL expression and/or activity are associated with obesity and IR [8]. This enzyme is regulated by various hormones including norepinephrine (NE), whose intracellular signaling is mediated by cyclic adenosine monophosphate that activates protein kinase A (PKA) [9]. It also regulates the perilipin A phosphorylation which allows HSL to translocate from the cytosol to the lipid droplets [10]. Alterations in LPL expression and/or activity can cause hypertriglyceridemia and obesity [11]. LPL hydrolyzes TG in circulating chylomicrons and VLDL generating NEFAs and glycerol, which are re-esterified and stored in adipocytes [12]. In addition, adipocytes secrete a variety of biologically active molecules called pro-inflammatory adipokines which participate in MS development [13]. In our laboratory, we have developed a MS model in Wistar rats that is induced by chronic administration of 30% sucrose in the drinking water for 20–24 weeks after weaning. The physiological abnormalities that are usually present in this model are intra-abdominal obesity, endothelial dysfunction, hypertriglyceridemia, high systolic blood pressure (SBP), hyperinsulinemia, IR, increase of NEFAs, renal failure, and fatty liver [14].

The purpose of this study was to evaluate the progressive participation of enzymes involved in lipolysis and lipogenesis that lead to adipocyte hypertrophy in intra-abdominal fat during the establishment of the MS in our rat model. Progressive changes in this model have been poorly addressed, since most research has only focused on changes at the end of the 20–24 weeks of 30% sucrose chronic treatment.

## 2. Materials and Methods

### 2.1. Animals and Diet

Experiments on animals were approved by the Laboratory Animal Care Committee of the National Institute of Cardiology “Ignacio Chávez” in México (protocol #17-1027) and were conducted in compliance with the Guide for the Care and use of laboratory animals of the NIH. A total of 70 male Wistar rats aged one month and weighing 100 ± 5 g were used (*n =* 7 male rats per group), and randomly separated into two groups: Group 1: 35 rats that consumed 30% sucrose in drinking water (MS) and Group 2: 35 control rats (C) that consumed water without sucrose. All animals received commercial food for rodents that contained crude protein 23%, crude fat 4.5%, ashes 8%, and added minerals 2.5% (Lab Diet 5008; Richmond, IN, USA). Each group was subdivided into five subgroups (subgroup MS1, MS2, MS3, MS4, MS5, and C1, C2, C3, C4, C5). Paired subgroups were sacrificed every month (from two months of age to six months of age), in such a way that, between the first and the last group, there was a difference of five months. Therefore, at the end of the experiments in the MS5 and C5 groups the rats were six months old.

The mean systolic blood pressure (SBP) in healthy rats has been reported to be 116 mmHg [15], and 112.8 mmHg with a variance of 19 in rats from the National Institute of Cardiology Ignacio Chavez. Based on this, the sample size per group was estimated by μ (SBP of Wistar rats) with 95% confidence and maximum error (ME) of 3.2 mmHg, according to the formula:$$\mathrm{ME}:\;\left|\mu -\bar{x}\right|=\;\left|116-112.8\right|=3.2\;\mathrm{mmHg}.$$
$$\mathrm{Confidence}:\;\left|\mu -\bar{x}\right|=\;\left|116-112.8\right|=3.2$$
$$\mathrm{SS}:\;n=\;\frac{{\left(\sigma_{\bar{x}}\right)}^{2}\sigma_{x}^{2}}{EM^{2}}=\;\frac{\left(2^{2}\right)\left(22\right)}{{\left(3.2\right)}^{2}}=\;\frac{4*19}{10.24}=\;\frac{76}{10.24}=7.42$$
where,
ME = Maximum error.$\sigma_{\bar{x}}$ = # of the standard deviations of the mean estimator.$\sigma_{x}^{2}$ = Variance of the SBP of the rats in our Institute.SS = sample size.

The animals were housed in ad hoc plastic boxes and were subjected to 12-h light/darkness cycles and the environmental temperature was kept between 18 and 26 °C. At the end of the experimental period, the rats were weighed and their SBP was measured by the tail-cuff method [16]. Animals were kept in fasting conditions for 12 h and weighed before sacrifice. The animals were subjected to euthanasia by decapitation and blood was collected in vacutainer tubes from the severed neck. The blood samples were centrifuged for 20 min at 936 g and 4 °C, in order to collect serum in aliquots of 500 μL and stored at −30 °C until used to quantify triglycerides (TG), glucose, total cholesterol (TC), insulin, advanced glycation end products (AGEs), and leptin.

### 2.2. Serum Biochemical Variables

Commercial kits were used for the determination of some serum biochemical variables in the rats; glucose concentration was determined by enzymatic SERA-PAKR Plus kit (Bayer Corporation, 108 S’ees, France). TC and TG determinations were made using commercial enzymatic kits, (RANDOX Laboratories Ltd., Crumlin, County Antrim, UK). Leptin and insulin were determined using commercial radioimmunoassay kits (RIA) (Linco Research Saint Charles. Inc., MI, USA). Advanced glycosylation products (AGEs) concentrations were determined according to a method previously described by Jiménez et al. [17]. The HOMA index of resistance to insulin was calculated. HOMA−IR = insulin μU/mL × glucose mM/L/22 5.

### 2.3. Adipocyte Isolation

The intra-abdominal fat was separated, weighed and the white adipocytes were isolated by collagenase digestion as described by Rodbell [18] with the modifications described by Guerra et al. [19]. In brief, 3 mg of type II collagenase dissolved in 10 mL of Krebs buffer containing 2% bovine serum albumin (BSA), NaCl 118 mM, NaHCO_3_ 24 mM, KH_2_PO_4_ 1.2 mM, MgSO_4_, 1.2 mM, Ca_2_Cl 4.7 mM, and D-glucose 4.5 mM at pH 7.35 were added to 3 g of adipose tissue. Subsequently, the tissue was cut mechanically with scissors and placed in gentle thermo-stirring at 37 °C for 90 min, then filtered through a cell dissociation sieve tissue grinder kit equipped with a 50-pore mesh and centrifuged for 1 min at 1500 rpm. The superficial dense layer was recovered and washed with 10 mL of Krebs buffer without BSA pH 7.35 and centrifuged for 1 min at 1500 rpm. This process was carried out twice and, subsequently, 900 μL of sucrose buffer were added containing EDTA 1 mM, TRIS 10 mM, sucrose 250 mM, 50 μL triton X-100, and 100 of protease inhibitor cocktail (PMSF 1 mM, pepstatin A 2 mM, leupeptin 2 mM and aprotinin 0.1%), vigorously stirred, homogenized and centrifuged for 5 min at 3500 rpm. The supernatant was recovered and frozen at −30 °C, until use. Total proteins were determined by the Bradford method [20].

### 2.4. Western Blotting for LPL, PKA, Perilipin A, and HSL

A total of 100 μg of protein from the adipocyte homogenate were separated by SDS-PAGE (10% polyacrylamide gel) and transferred to a nitrocellulose Hybond-P membrane 45 µM (Millipore). The blots were blocked for 3 h at room temperature with Tris-buffer solution (TBS), containing nonfat dry milk 5% and Tween 20, 0.05% pH 7.4. The membranes were incubated overnight at 4 °C with rabbit primary polyclonal antibodies against LPL, perilipin A, and mouse monoclonal against PKA and HSL at a 1:1000 dilution (Santa Cruz Biotechnology, Santa Cruz, CA, USA and abcam respectively), with constant shaking. The blots were washed with TBS and incubated for 3 h at room temperature with secondary polyclonal antibodies, and peroxidase-labeled, (Santa Cruz Biotechnology, Santa Cruz, CA, USA) at a final dilution of 1:1200. After incubation with the secondary antibody, the membranes were washed with TBS, and revealed using 3, 3′-diaminobenzidine. After identifying the relevant protein, the membranes were stripped by washing with a buffer of Tris 1%, SDS 1%, and β-mercaptoethanol 100 mmol/L, pH 2 for 2 h, followed by a washing with TBS plus NaCl 5%. The membranes were blocked for 3 h and incubated overnight with a mouse monoclonal biotinylated glyceraldehyde-3-phosphate dehydrogenase (GAPDH) antibody (1:2000). The membranes were analyzed by densitometry using 1D Kodak Image analysis software, Windows Version 3.5.

### 2.5. Retroperitoneal Fat Histology

For histology, 250 mg of retroperitoneal fat were washed in NaCl 0.9% for 30 sec. The solution was then decanted and phosphate buffer pH 7.35 with formalin 10% was added for 24 h. The histological sections were processed according to conventional histological procedures and stained with Masson technique. Histological sections were analyzed using a light microscope Carl Zeiss (63300 model) equipped with a Tucsen (9 megapixels) digital camera with software TSview 7.1, at a 40× magnification. The intensity of light in the microscope was adjusted and remained constant. The photomicrographs were analyzed by densitometry using Sigma Scan Pro 5 Image Analysis software (Inc. San Jose, California, CA, USA), and parameters of analyses in the software were adjusted and remained constant for all groups. An average of five histological sections were examined. The density values are expressed as µm. Fat cell number per gram of tissue was calculated using the method described by Commerford et al. [21].

### 2.6. LPL and HSL Activities

LPL activity was determined by the rate of hydrolysis of emulsified triolein substrates by endogenous LPL. Triolein was emulsified in arabic gum 5% and free fatty acid 2% BSA. Then, 100 μg of homogenized adipocyte solution per sample were added to 100 μL of triolein 5 mM substrate solution, 100 μL of Tris-HCl 0.2 M buffer, and 0.2% Triton X-100, pH 7.4 (reaction A). Homogenized adipocyte solution without triolein substrates (reaction B) was run in parallel. The reactions were incubated 1 h at 37 °C in a shaking water bath. At the end of the incubation period, when the endogenous LPL had hydrolyzed triolein to release glycerol, 50 μL of NaCl 1 M were added. Free glycerol concentration in the samples was determined by the use of a colorimetric assay kit (GPO Trinder reaction) from the absorption at 540 nm. The calculated A-B value in the samples indicated the rate of glycerol release by LPL and represented the LPL activity. Lipolysis data are expressed as nmol of glycerol per mg protein.

The HSL activity was determined by the hydrolysis rate of the endogenous TG of the adipocytes. A total of 100 μg of adipose tissue were incubated in parallel; sample A: contained 100 μg of adipose tissue plus 100 of isoproterenol 100 nM and 900 μL of Krebs buffer pH 7.35, Sample B: contained 100 μg of adipose tissue plus 1 mL of Krebs buffer, pH 7.35, both with constant agitation, at 37 °C, for 60 min in a shaking water bath. At the end of the incubation period, 50 μL of cold NaCl 1 M were added to stop the reaction. Free glycerol concentration in the samples was determined by the use of a colorimetric assay kit GPO and the absorbance was read at 540 nm. The value obtained from the sample A minus that of sample B in the samples indicated the rate of glycerol release by HSL and represents the HSL activity. The data are expressed as nmol of free glycerol per g of tissue.

### 2.7. Extraction and Derivatization Non-Esterified Fatty Acid (NEFAs)

A total of 5 mg of homogenized adipocyte solution were used in the presence of 100 μg of nonadecanoic acid as internal standard, 2 mL of chloroform methanol (2:1, vol/vol) with 0.002% BHT, as was described by Folch [22]. The obtained lipid residue was dissolved at room temperature for 15 min in 1 mL of methanol containing 100 of 2, 2-dimethoxypropane and 10 μL of concentrated H_2_SO_4_ to esterifies NEFAs to their corresponding methyl esters as described by Tserng [23]. These reaction conditions are necessary to avoid the esterification of FA from phospholipids, cholesterol esters, and TG. The concentration and composition of NEFAs methyl esters were separated and identified by gas chromatography-FID in a Carlo Erba Fratovap 2300 chromatograph equipped with a capillary column packed with the SP-2330 phase (30 m long and 0.25 mm 0.2 mm film thickness) and fitted with a flame ionization detector at 210 °C, with helium as the carrier gas at a flow rate of 1.2 mL/min.

### 2.8. Statistical Analysis

Statistical analyses and graphs were performed with the SigmaPlot 14 program, Jendel Corporation, 1986–2017. The data are presented as the mean ± SEM. Statistical significance was determined by one-way ANOVA test, followed by Tukey’s post hoc test. Differences were considered as statistically significant when *p* ≤ 0.05.

## 3. Results

### 3.1. General Characteristics

The body weight, water intake, and ingested food in the C groups of each month did not show a significant difference in comparison with their paired groups of MS except for a tendency to gain weight in the group MS5 that did not reach a statistically significant difference. Only ingested food in the rats of the MS5 group showed an increase in comparison to C5 (*p =* 0.001, Table 1). The SBP in the first month did not show a significant difference between C and MS groups. However, in the second, third, fourth, and fifth months a significant increase was present in MS groups vs. their C groups (*p =* 0.001, Table 1). We did not find significant differences in the TC and glucose concentrations in any of our experimental groups.

Serum TG levels were significantly lower in all C groups (*p =* 0.001, Table 2), when compared to the corresponding MS groups. Insulin serum concentration and HOMA-index in the first and second month did not show a significant difference between C and MS groups. However, during the third, fourth, and fifth months, they showed significant increases in MS in comparison to C groups (*p =* 0.001, Table 2). The leptin serum concentration showed no significant difference between the C vs. MS group in the first month. However, the leptin concentration gradually and significantly increased from the second to the fifth month vs. the MS groups (*p =* 0.001, Table 2). The AGEs concentration in serum did not show a significant difference between C and MS groups in the first, second, and third months. However, in the fourth and fifth month there was a significant difference in the MS rats when compared to their paired C groups (*p =* 0.01, Table 2).

### 3.2. NEFAs Composition in Intra-Abdominal Adipocyte Homogenate

NEFAs composition in the first and second month did not show significant differences between the C and MS groups. However, from the third month and until the fifth month, there was a significant increase in palmitic, palmitoleic, and oleic fatty acids in MS rats in comparison with the rats of the C groups (*p* < 0.04). Linoleic acid in MS rats of the fourth and fifth month showed significant decreases vs. their paired C groups, respectively (*p* < 0.04, Table 3).

### 3.3. Intra-Abdominal Adipocytes

Figure 1A shows that the weight of intra-abdominal fat of the first and second month showed no significant differences between C and MS groups. However, from the third, and until the fifth month, there were significant increases between the MS and C groups (*p =* 0.001). Figure 1B shows that the fat cell number per gram of tissue showed no significant differences during the first and second month between C and MS groups. However, from the third month, and until the fifth month, there were significant increases between the MS and C groups (*p =* 0.02 and *p =* 0.01, respectively). A decrease in adipocyte cell number per field of the MS rats in comparison to their paired C groups was observed (*p =* 0.001 and *p =* 0.002 respectively, Figure 1C). These increases were reflected in adipocyte diameter area from the third month and until the fifth month in comparison to their paired C groups (*p =* 0.01, *p =* 0.001 and *p =* 0.005, respectively, Figure 1D).

Figure 2 shows representative photomicrographs of adipose tissue, obtained from intra-abdominal fat tissue samples of the rats in C and MS groups. Figure 2A–E represents the C groups corresponding to the successive months of treatment (for example, Figure 2A and F = 1 month). Adipocytes have membrane-bound cytoplasm, the cell nucleus is oriented to the periphery, and the cells have a quasi-hexagonal shape. Figure 2F–J shows representative microphotographs of intra-abdominal adipose tissue from the MS groups and, in them, there are oval shaped adipocytes with the cell nucleus totally oriented to the periphery. There is a progressive increase in the circumference of the adipocytes. In addition, in Figure 2H–J, the space of the cytoplasm is observed with little clarity. These histological results were associated with the amount of intra-abdominal fat, adipocyte cell number per field, the fat cell number per gram of tissue, and adipocyte diameter area. They clearly show adipocyte hypertrophy.

### 3.4. LPL and HSL Activities in Adipocyte Homogenate

Figure 3A shows that the LPL activity showed no significant changes in the first and second months when comparing the MS vs. C groups. However, from the third month on, there was a tendency to increase that became statistically significant in the fourth and fifth months (*p =* 0.03 and *p =* 0.01, respectively). This same tendency was observed in the HSL activity (*p =* 0.03, *p =* 0.01, and *p =* 0.02 respectively, Figure 3B).

### 3.5. PKA, Perilipin A, HSL, and LPL Expressions

Figure 4A shows that in the MS and C groups of the first month there was no significant difference in the PKA expression. However, from the second month and until fifth month there were significant increases in MS groups compared to their paired C groups (*p =* 0.03 and *p =* 0.05, respectively). Figure 4B shows that there were no significant increases between the MS and C groups in the first and second months in the perilipin A expression. However, from the third month and until the fifth month, there was a significant decrease in the MS groups in comparison with their paired C groups (*p =* 0.03, *p =* 0.05, and *p =* 0.01, respectively). Figure 5A shows a significant difference between HSL expression in the MS groups with the paired C groups from the first month and until the fifth month (*p =* 0.04, *p* < 0.001, *p* < 0.001, *p =* 0.01, and *p =* 0.02, respectively). Figure 5B shows that there were no significant differences between the LPL expressions in any groups throughout the five months studied in the MS rats when compared with the paired C groups.

## 4. Discussion

The aim of this study was to evaluate how chronic consumption of 30% sucrose progressively alters the main enzymes that participate in lipolysis and lipogenesis, which may contribute to adipocyte hypertrophy in a MS rat model. Progressive changes in time have been poorly addressed in this model, since most of the research has only focused on changes in obesity, hyperinsulinemia, and IR at the end of the 20–24 weeks [24].

### 4.1. Progressive Changes in Body and Biochemical Variables during the Establishment of the MS

Our results indicate that the MS model shows a significant progressive increase in intra-abdominal fat and SBP since the third month when compared to the C groups. We did not find significant differences in TC and glucose concentrations in any of the MS experimental groups. Therefore, this model has little alterations in glucose metabolism [16] and does not develop diabetes. This fact has been previously described by Reaven and Ho [25]. There is also a report from another study which described that in MS patients, normal serum glycemic values were present [26]. However, the consumption of the hyper-caloric sucrose solution by MS rats provides high amounts of calories that lead to lipogenesis and favor intra-abdominal fat deposits [27]. A study in rats showed that 30% sucrose administration in drinking water for 21 weeks leads to MS development in male Wistar rats with an increase in body weight, SBP, insulin, TG, LDL lipoproteins, and NEFAs [28]. Our present results show that TG, insulin, SBP, and AGEs are gradually increased month by month in the MS groups when compared to the C groups, but without changes in body weight, despite the tendency to increase body weight in the MS group of five months which was not statistically significant.

The HOMA-IR index was used as an IR indicator. The usefulness of this method was shown by a study conducted in healthy subjects without differences in sex, age or pathology, in which its use reflected insulin sensitivity [27]. A significant increase in HOMA-IR in a MS model in which fructose (20–25%) was given for 9 weeks has also been found [29]. Our results show a significant increase in the HOMA-IR index elicited by high sucrose consumption from the third month on in the MS rat model. During IR, insulin action at the cellular level is reduced in several tissues, which increases the secretion of this hormone by the pancreas [30].

In addition, the HOMA-IR values are closely related to the concentration of AGEs [31], which increase due to the chronic hypercaloric consumption. This consumption causes oxidative stress and is associated with IR [32]. AGEs can induce structural and functional damage to receptors present in the cell membrane, modify structural enzymes leading to alterations in their catalytic ability, and cause the formation of cross-links with proteins of the extracellular matrix. The insulin receptor may be altered by AGEs. This receptor has been associated with sarcopenia in diabetes. [33].

### 4.2. Changes in Abdominal Tissue

Despite the tendency to increase in body weight in the MS group of five months, there was no significant change in body weight. However, a progressive increase in intra-abdominal fat was observed in the MS groups from the third month on, when compared to C rats. The absence of an increase in body weight despite the increase in intra-abdominal fat could be due to a decrease in muscle mass. This may be due to a diminished consumption of solid food in the MS rats, which results in a decreased protein and mineral intake in spite of a higher calorie intake. Therefore, rats probably develop sarcopenia [34] which may be the result of increased circulating NEFAs that can be deposited in muscle [35]. However, this hypothesis requires future research. Furthermore, the muscular lipid accumulation impairs insulin action and can induce activation of MAPK-mediated IR and this could cause muscle wasting and a decrease of IGF-1 [34]. In addition, the excess of NEFAs from the intra-abdominal adipose tissue could deposit in the liver facilitating IR development and hepatic steatosis [27], thus creating a positive feedback process between the liver, pancreas, and intra-abdominal adipose tissue.

Intra-abdominal obesity is characterized by elevated rates of synthesis and release of adipokines, such as leptin, which is an anorexigenic hormone. During obesity, leptin resistance (LR) is developed, and this is directly related to IR [36]. DS/obese rats have a high serum TG concentration, hyperinsulinemia, and LR, despite being fed normal diets. This strain of rats shows similarities to humans that develop MS [37]. In normal conditions, leptin suppresses insulin secretion through central actions and through direct effects on pancreatic beta cells resulting in the reduction of insulin levels. However, in obesity, this signaling is altered causing LR which promotes obesity, hyperinsulinemia, and IR [38]. Circulating levels of leptin are proportional to the amount of body fat, and therefore, there is hyperphagia despite there being leptin excess, leading to LR [39]. Leptin production is directly proportional to increases in adipocyte size, representing a self-regulation mechanism. However, when LR appears, it suppresses cytokine 3 signaling leading to an imbalance in this metabolic pathway [40].

### 4.3. Lipogenesis and LPL

The LPL activity regulates TG accumulation in adipocytes. In obese MS Zucker rats, the activity of this enzyme is elevated and is associated with an increase in hepatic lipogenesis and hyperinsulinemia [41]. Our results show that the LPL activity was progressively increased from the third month on, which suggests that the chronic consumption of carbohydrate induces a progressive elevation in the activity of this enzyme, favoring adipocyte hypertrophy. Hypertrophy is associated with hyperinsulinemia and IR [41,42].

The LPL activity in intra-abdominal adipocytes is increased, favoring the fat net deposition that leads to the development of hypertrophic intra-abdominal obesity in the MS rats [40]. However, the LPL expression remains unchanged which suggests that the expression of this enzyme can be inversely related to its activity [24]. These results are similar to those described by Cheng et al. who found that there was adipocyte hypertrophy, obesity, and hypertension in adult male Sprague Dawley rats that developed MS after ingesting a hyper-caloric diet [43]. Another study demonstrated that LPL is overexpressed in MS subjects in comparison with healthy subjects [44]. In addition, several studies in animal models and humans suggest that hyperinsulinemia can over-stimulate the catecholamine pathway and that NE inhibits LPL in specific tissues. NE negatively regulates LPL in skeletal muscle in hyperinsulinemic states, while it positively regulates it in adipose tissue [45]. In our model, there is a significant increase in NE dependent vasoconstriction [46], which suggests that the catecholamine pathway could contribute to increase the LPL activity.

The 30% sucrose chronic consumption of sucrose in this study caused an increase in lipogenesis through the elevation of the LPL activity, which is constantly stimulated by changes in the liver metabolism which increase TG synthesis and NEFAs re-esterification. It has also been demonstrated that the albumin glycation by AGEs increases TNF-α production, which is related to IR by inducing a pro-inflammatory processes that suppresses the transduction of the insulin signaling in the adipocyte, causing an alteration in the LPL activity [31].

### 4.4. Lipolysis and HSL

Our results show that the HSL activity and/or expression are gradually increased from the third month on, and this could be due to its stimulation by the chronic hyper-caloric insult of sucrose. A previous study demonstrated that there was significant increase in the HSL activity the presence of IR and hyperinsulinemia in female Fischer rats receiving a high calorie diet intake [42]. Our results agree with these results and suggest that the HSL activity is determined in part by the adipocyte size.

It has been described that both activity and expression of HSL may be directly associated. However, the lipolytic capacity of this enzyme is usually related to its activity and not its expression [47], due to the fact that the expressed enzyme is not always functional. Therefore, the enzymatic activity seems to be a more accurate measurement of its effect. Several studies have described that during IR, HSL expression is not modified while its activity is increased by constant phosphorylation of the enzyme [48]. Other studies have reported that the HSL expression is not altered but its activity is increased in cellular cultures of adipocytes from murine obesity models induced by diet [49]. These results might seem paradoxical, since a decrease in the adipocyte size associated with the increase the HSL activity and/or over-expression, and NEFAs liberation with the circulation, would be expected and is not always reported.

The rate of lipolysis has been associated to hyperglycemia and hyperinsulinemia which are alterations frequently present in obesity and type-2 diabetes [50]. A previous study showed a high HSL expression in cells from pancreatic islets of rats exposed to high glucose concentrations. This induction is parallel to an increase in lipolysis and is accompanied by an increase in the HSL activity, the amount of protein, and the mRNA level of this enzyme, suggesting that this effect is mediated by the transcription of the gene that codes for HSL [51]. In addition, the continuous exposure of isolated adipocytes to high glucose and insulin concentrations increases the lipolysis rate and the expression of HSL [50]. Our results show that in our MS rats, the chronic sucrose consumption does not change the serum glucose concentration, but significantly increases the serum insulin concentration from the second month on. This was found to be associated with changes in HSL activity and/or expression. An elevated lipolytic activity in adipocyte is associated with obesity and an increase in IR and NEFAs. This could probably be due to the presence of a constantly phosphorylated HSL that would induce an increase in the lipolytic rate [24,50].

Leptin gene overexpression in adipocyte and elevation of circulating leptin levels can also contribute to enhanced basal lipolysis in obesity [52] due to its action on the leptin receptor [53]. Increased leptin circulation may enhance lipolysis by counteracting the antilipolytic effects of insulin which include its ability to inhibit the β-adrenergic receptor-mediated lipolysis and activation of PKA [54]. Furthermore, a significant increase in leptin has been associated with elevated TG concentrations [55]. In this study we found that leptin and TG were significantly increased from the first and second months respectively. Furthermore, in obese subjects, IR causes an increase in intracellular TG hydrolysis by adipocytes, and consequently, liberation of NEFAs to the circulation. Here we demonstrated that our model is characterized by a progressive increase in serum NEFAs through the months of treatment with 30% sucrose. The excess of NEFAs in the circulation are associated with endothelial damage and elevation of SBP which contributes to hypertension by inactivating the eNOS pathway [56,57].

HSL phosphorylation on multiple sites activates it, leading to the subsequent translocation of the lipase from the cytosol to the lipid droplet. This step is catalyzed by activated PKA [53]. Leptin directly activates PKA together with catecholamines. The β-adrenergic receptor is highly expressed in adipose tissue and NE binds to the adipocyte cell membrane [45,58]. These receptors are coupled to G-proteins that transmit a stimulatory signal to adenylyl cyclase to generate cyclic AMP. Cyclic AMP, in turn, binds to PKA, causing the regulatory subunits to dissociate from the catalytically active subunits, resulting in an increased activity of the enzyme [59]. Insulin promotes the coupling and phosphorylation of β-adrenergic receptors in the adipocyte cell membrane of C57BL/6J male rats with hypercaloric diet, thus promoting PKA phosphorylation. Our results show that PKA expression significantly increased from the second month on in MS rats [60]. Treatment with a PKA inhibitor, in Sprague-Dawley rats fed with hypercaloric diet, regulates the transcription the LPL genes and improves IR [61]. Another study showed that after exposure to a hypercaloric diet in knockout mice deficient in one of the main PKA regulatory subunits in adipose tissue, resulted in obesity, IR, and hepatic steatosis, in comparison with wild mice [62].

Perilipin A which is expressed in mature adipocytes also phosphorylates HSL and promotes its entry into the lipid droplet [60]. Under un-stimulated conditions, perilipin A coating the lipid droplet acts as barrier that restricts access of HSL to TG substrates in order to prevent unrestrained basal lipolysis [63]. Perilipin A is an important enzyme-regulated switch that controls lipolytic rate and hence, efflux of NEFAs from adipose tissue [64]. An increase in PKA is associated with a decrease of perilipin A and an increase in HSL. Perilipin A phosphorylation by PKA may result in an increase in the activity of multiple lipid droplet associated lipases, such as desnutrin/ATGL and HSL [53]. Our results show a decrease in the expression of perilipin A from the third month on in the MS groups. This suggests that perilipin A is a key enzyme which is altered in the adipocyte in MS rats [24]. The decrease in perilipin A expression in the lipid droplet results in the access of HSL to TG stored in the lipid droplet. A decrease of perilipin A in obese women, and in the intra-abdominal adipose tissue from obese subjects, has been reported. This decrease is associated with a decrease of the mRNA of this enzyme [65]. Perilipin A expression is reduced in obese women, when compared to that in lean women, and this is associated with an increase basal lipolysis [66]. Another study, in ovariectomized diabetic female rats, showed an increase in adipose tissue through the reduction of perilipin A expression which was associated with hyperinsulinemia and IR [67]. Our results suggest that the decrease in the perilipin A expression in MS rats could be associated with an increase in PKA and HSL activity and/or expression, thus increasing the basal lipolysis that can be reflected in an increase in NEFAs [68]. Besides, another study demonstrated that by giving a hyper-caloric diet to Sprague-Dawley rats there was an increase in the expression of perilipin A, with the consequent increase in serum insulin and leptin [69]. However, another study demonstrated that when giving a hyper-caloric diet to female rats and male knockout rats to perilipin, there was resistance to obesity and low concentrations of NEFAs, TC, and TG, but intolerance to glucose with peripheral IR. These results were associated with increased lipolysis in the adipocytes of the knockout rats which suggested the loss of the protective action of phosphorylated perilipin on the surface of the lipid droplet promoting the expression and/or activity of HSL [64].

In summary, the 30% sucrose chronic consumption caused an increase in lipogenesis through the elevation of the LPL activity, which is constantly stimulated by changes in the liver metabolism which increase TG synthesis and NEFAs re-esterification. In addition, the HSL activity and/or expression is associated with hyperinsulinemia that increases the NEFAs release causing a positive feedback between the intra-abdominal fat, liver, and pancreas thus significantly contributing to the progression of MS. Additionally, PKA and perilipin A which regulate HSL activity show progressive changes during the establishment of the MS. Therefore, there is a close association between the lipogenesis and the lipolysis processes, and both are altered in the adipocytes of the MS rats since the third month, leading to adipocyte hypertrophy in intra-abdominal fat. Hypertrophy is evidenced by an increased adipocyte area and a decreased cell number per field.

## 5. Conclusions

In a MS model caused by chronic consumption of 30% sucrose for five months, the enzymes involved in lipolysis and lipogenesis in the adipocyte of the intra-abdominal tissue are progressively altered. The HSL, LPL, and PKA expressions and/or activities are increased and perilipin A is decreased. All changes are progressive from the third month on. Changes of these enzymes were associated with progressive adipocyte hypertrophy and NEFAs causing a positive feedback that contributes to the development of adipocyte hypertrophy.

## Figures and Tables

**Figure 1 nutrients-11-01529-f001:**
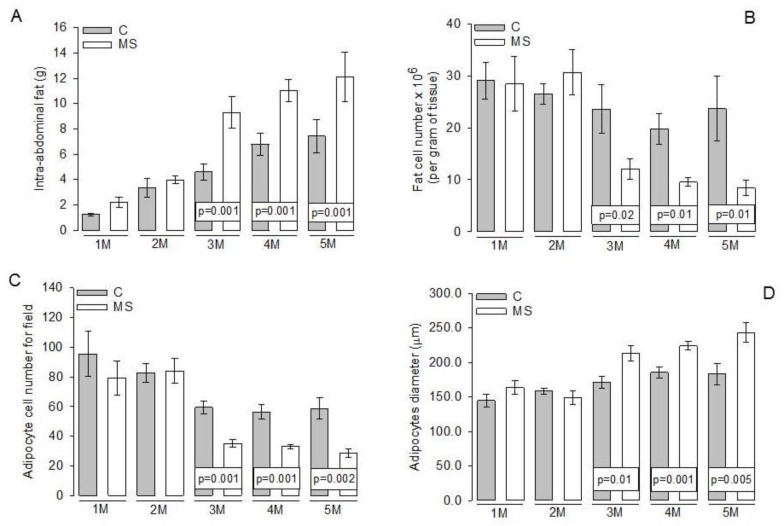
Representative histograms: (**A**) intra-abdominal fat. (**B**) Fat cell number per gram of tissue. (**C**) Adipocyte cell number for field and (**D**) adipocyte diameter. Data show the mean ± SEM; *n* = 7 rats per experimental group. Abbreviations: C = Control; MS = Metabolic syndrome; M = Month.

**Figure 2 nutrients-11-01529-f002:**
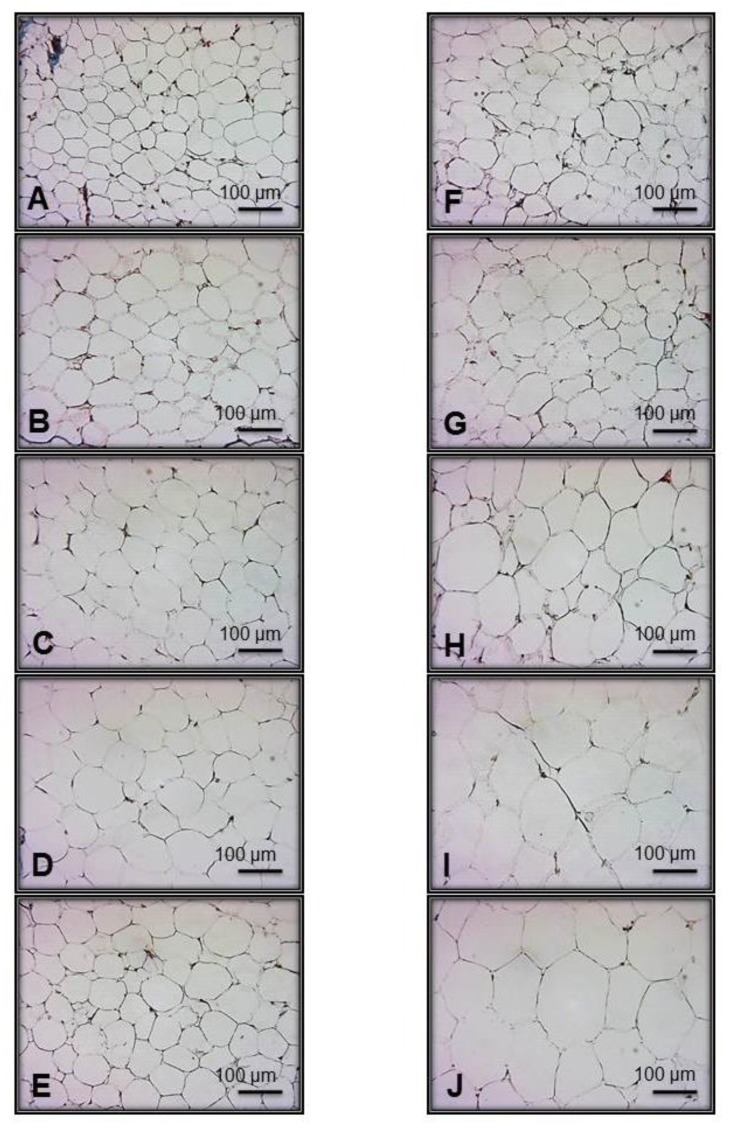
Representative photomicrographs of intra-abdominal white tissue from the experimental groups that show adipocyte size. Five fields per sample were analyzed. Panels (**A**) 1 M, (**B**) 2 M, (**C**) 3 M, (**D**) 4 M and (**E**) 5 M represent the C, and panels (**F**) 1 M, (**G**) 2 M, (**H**) 3 M, (**I**) 4 M and (**J**) 5 M represent the MS groups. *n* = 7 rats per experimental group. Abbreviations: C = Control; MS = Metabolic syndrome; M = Month.

**Figure 3 nutrients-11-01529-f003:**
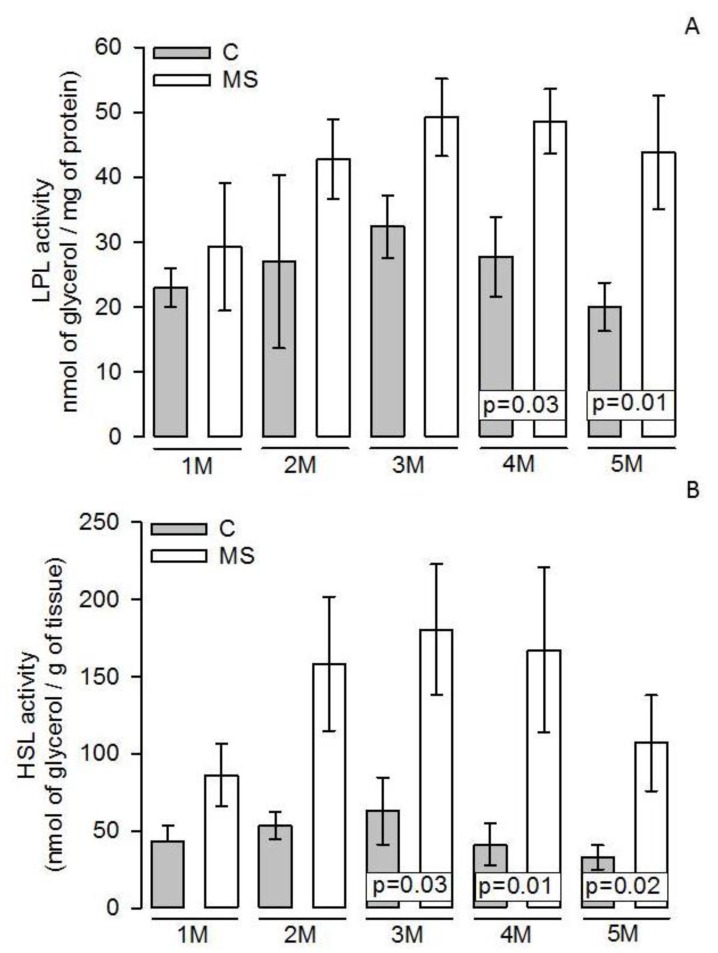
Effect of 30% chronic sucrose on the LPL (**A**) and HSL (**B**) activities through the five months of the treatment determined in the adipocyte homogenates in the experimental groups. Values are the mean ± SEM, *n* = 7 rats per experimental group. Abbreviations: C = Control; MS = Metabolic syndrome; LPL = Lipoprotein lipase; M = Month; HSL = Hormone sensitive lipase.

**Figure 4 nutrients-11-01529-f004:**
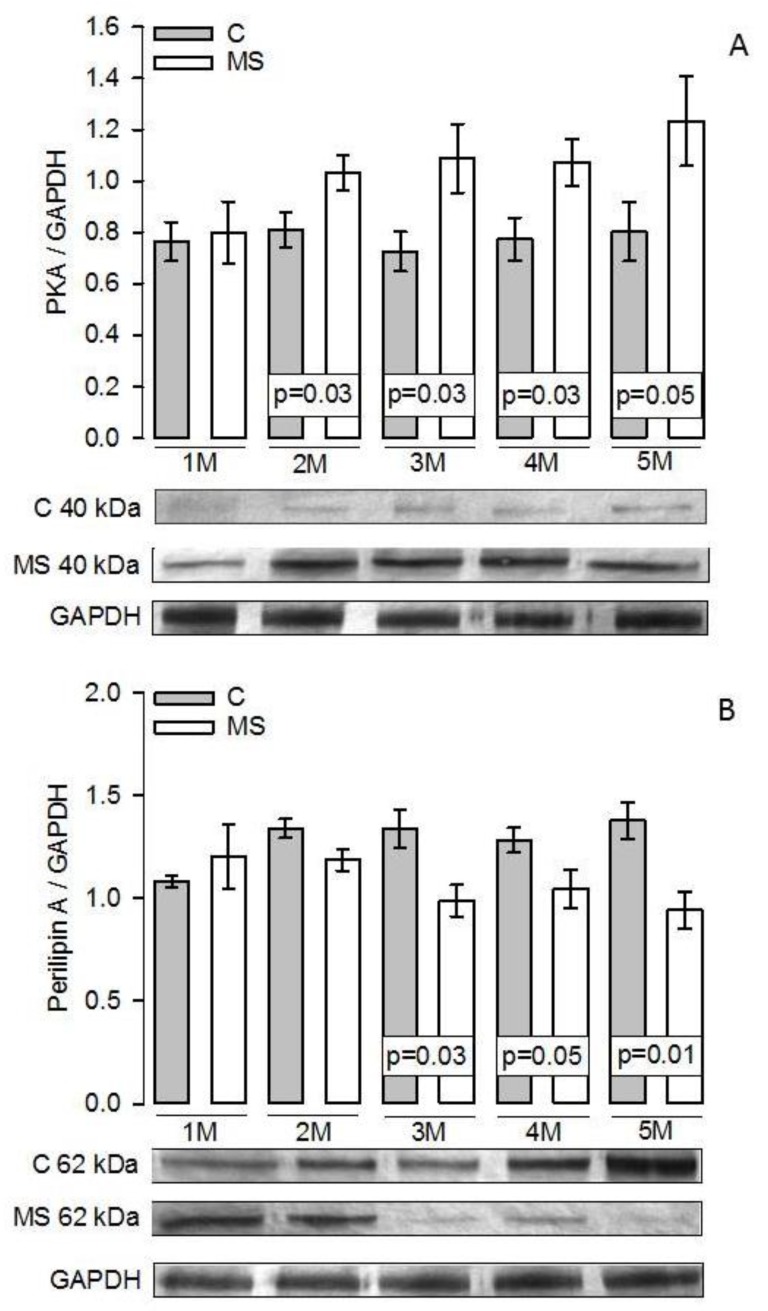
Representative histograms of the PKA/GAPDH expression (**A**) and Perilipin A/GAPDH expression (**B**). Values are the mean ± SEM *n* = 7 rats per experimental group. Abbreviations: C = Control; MS = Metabolic syndrome; PKA = Protein kinase A; GAPDH = Glyceraldehyde-3-phosphate dehydrogenase; M = Month.

**Figure 5 nutrients-11-01529-f005:**
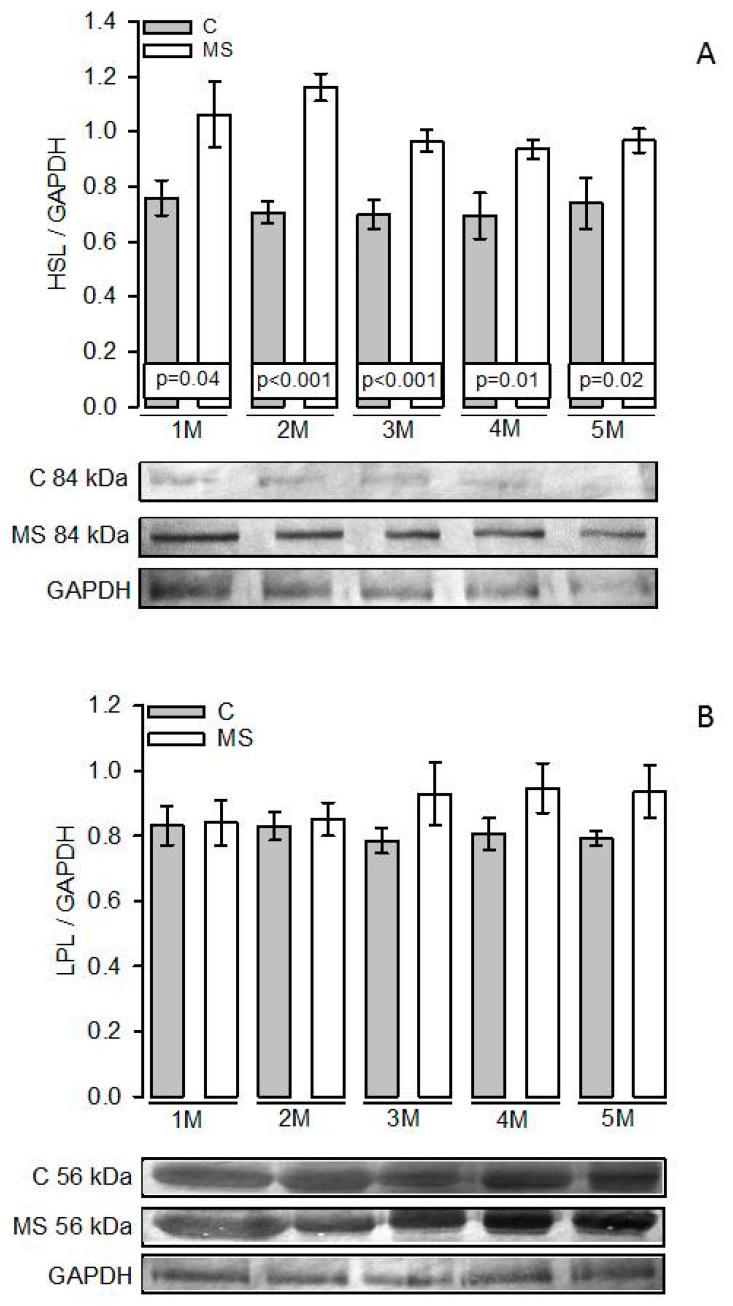
Representative histograms of HSL/GAPDH expression (**A**) and LPL/GAPDH expression (**B**). Values are the mean ± SEM, *n* = 7 rats per experimental group. Abbreviations: C = Control; MS = Metabolic syndrome; LPL = Lipoprotein lipase; M = Month; HSL = Hormone sensitive lipase.

**Table 1 nutrients-11-01529-t001:** General characteristic in experimental groups.

Variables	1 Month		2 Months		3 Months		4 Months		5 Months	
	C	MS	C	MS	C	MS	C	MS	C	MS
Body weight (g)	228.2 ± 11.1	228.2 ± 17.1	317.2 ± 12.3	302.1 ± 16.7	413.7 ± 16.3	422.0 ± 17.2	491.3 ± 22.4	477.1 ± 21.5	443.0 ± 23.6	492.6 ± 13.4 ^§^
Water Intake (mL)	27.5 ± 1.0	15.7 ± 0.4	24.7 ± 1.6	13.2 ± 1.2	21.7 ± 1.8	29.8 ± 0.8	29.0 ± 1.5	17.2 ± 4.8	11.4 ± 1.7	19.1 ± 1.6
Ingested food (g)	37.1 ± 1.4	26.4 ± 1.4	39.2 ± 1.7	41.0 ± 1.1	53.5 ± 4.4	46.4 ± 1.7	50.7 ± 2.7	50.7 ± 4.5	45.0 ± 4.2	78.5 ± 6.4 ^†^
SBP (mmHg)	120.4 ± 2.9	125.9 ± 1.3	128.1 ± 1.4	136.9 ± 1.5 ^†^	121.5 ± 0.4	135.6 ± 0.4 ^†^	125.2 ± 0.2	139.4 ± 0.5 ^†^	125.2 ± 0.8	136.3 ± 0.2 ^†^

Data show the mean ± SEM, *n* = 7. Abbreviations: C = Control; MS = Metabolic syndrome; SBP = systolic blood pressure; NS = not significant. ^†^ C vs. MS *p =* 0.001, and ^§^ NS = 0.08.

**Table 2 nutrients-11-01529-t002:** Biochemical blood characteristics in experimental groups.

Variables	1 Month		2 Months		3 Months		4 Months		5 Months	
	C	MS	C	MS	C	MS	C	MS	C	MS
Glucose (mmol/L)	5.3 ± 0.2	5.6 ± 0.5	6.1 ± 0.3	5.7 ± 0.3	6.0 ± 0.1	5.9 ± 0.2	5.5 ± 0.1	5.7 ± 0.1	5.8 ± 0.2	5.5 ± 0.1
TC (mm/dL)	55.0 ± 3.0	59.2 ± 2.2	55.5 ± 2.7	52.2 ± 1.9	53.3 ± 1.7	50.7 ± 2.9	52.0 ± 1.4	53.8 ± 2.3	56.0 ± 2.7	47.5 ± 3.3
TG (mg/dL)	51.7 ± 2.1	110.1 ± 16.6 ^†^	55.5 ± 3.7	127.2 ± 14.5 ^†^	52.6 ± 2.0	100.0 ± 9.6 ^†^	44.8 ± 2.2	114.3 ± 7.0 ^†^	62.7 ± 4.4	94.7 ± 14.5 ^†^
Insulin (µU/mL)	1.5 ± 0.1	1.3 ± 0.3	2.0 ± 0.3	3.0 ± 0.4 ^†^	2.6 ± 0.1	4.1 ± 0.2 ^†^	2.0 ± 0.2	3.5 ± 0.3 ^†^	1.7 ± 0.3	4.0 ± 0.5 ^†^
HOMA IR	0.3 ± 0.04	0.3 ± 0.1	0.5 ± 0.06	0.6 ± 0.1	0.6 ± 0.05	1.0 ± 0.06 ^†^	0.5 ± 0.06	0.9 ± 0.09 ^†^	0.4 ± 0.08	1.1 ± 0.2 ^†^
Leptin (ng/mL)	4.2 ± 0.4	4.7 ± 0.5	5.7 ± 0.4	8.0 ± 0.4 ^†^	4.4 ± 0.6	8.8 ± 1.1 ^†^	3.9 ± 0.9	9.7 ± 1.4 ^†^	6.7 ± 1.5	12.7 ± 2.0 ^†^
AGEs (mU/mL)	235.5 ± 5.0	282.4 ± 26.4	259.5 ± 22.3	262.5 ± 74.1	185.5 ± 11.3	227.8 ± 18.1	145.1 ± 13.3	218.4 ± 30.4 *	162.4 ± 26.3	262.7 ± 30.7 *

Data show the mean ± SEM, *n* = 7. Abbreviations: C = Control; MS = Metabolic syndrome; CT = Cholesterol; TG = Triglycerides; AGEs = advanced glycation end products. * C vs. MS *p =* 0.01, ^†^ C vs. MS *p =* 0.001.

**Table 3 nutrients-11-01529-t003:** Non-esterified fatty acid (NEFAs) composition in intra-abdominal adipocyte homogenate of the experimental groups.

NEFAs (%)	1 Month		2 Months		3 Months		4 Months		5 Months	
	C	MS	C	MS	C	MS	C	MS	C	MS
Palmitic	28.4 ± 0.4	25.1 ± 3.4	28.2 ± 0.6	26.7 ± 0.9	24.5 ± 0.1	32.5 ± 1.1 ^†^	30.1 ± 0.8	34.2 ± 1.3 *	27.8 ± 1.5	34.1 ± 1.7 *
Palmitoleic	4.2 ± 0.2	4.9 ± 0.7	4.8 ± 0.5	5.8 ± 1.5	3.8 ± 0.3	13.7 ± 0.1 ^†^	5.8 ± 1.0	12.1 ± 0.9 ^†^	6.5 ± 1.3	12.6 ± 0.8 ^†^
Stearic	7.2 ± 0.1	4.8 ± 0.4	6.1 ± 0.4	4.6 ± 0.8	15.1 ± 7.5	6.8 ± 2.1	9.1 ± 2.0	12.4 ± 1.2	12.1 ± 1.8	10.1 ± 1.9
Oleic	38.7 ± 2.4	36.8 ± 3.8	37.1 ± 0.9	44.6 ± 0.9	30.1 ± 2.3	40.9 ± 2.3 *	31.7 ± 2.8	37.4 ± 2.2 *	29.0 ± 4.1	39.7 ± 2.4 *
Linoleic	17.9 ± 2.3	11.6 ± 0.9	22.2 ± 0.5	16.5 ± 2.3	18.3 ± 6.1	7.7 ± 2.3	14.7 ± 2.0	8.2 ± 1.6 *	19.1 ± 1.9	6.0 ± 1.5 ^†^
Total	93.4 ± 5.4	83.2 ± 9.2	98.4 ± 2.9	98.2 ± 6.4	91.8 ± 16.3	101.6 ± 7.9	91.4 ± 8.6	104.3 ± 7.2	94.5 ± 10.6	102.5 ± 8.3

Data are means ± SE. *n* = 7 different rats per group. Abbreviations: C = Control; MS = Metabolic syndrome; NEFAs = non-esterified fatty acid. Significantly different from C vs. MS * *p* < 0.04, † *p* = 0.001.

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
