# Peer review of "Intra-Abdominal Fat Adipocyte Hypertrophy through a Progressive Alteration of Lipolysis and Lipogenesis in Metabolic Syndrome Rats"

_nutrients, 2019, doi:10.3390/nu11071529_

Round 1
Reviewer 1 Report
In their manuscript entitled “Progressive alteration of enzymes involved in lipolysis and lipogenesis that lead to adipocyte hypertrophy in intra-abdominal fat in metabolic syndrome rats” by Perez-Torres et al., the authors investigated how activity and expression of enzymes involved in the lipolysis and lipogenesis are changed in time, using a metabolic syndrome rat model on 30% sucrose in drinking water diet. The present study is focused on progressive changes in some characteristics of the metabolic syndrome, making it an observational study with no proposed mechanism involved. Overall, I think this may contribute to our knowledge about this particular metabolic syndrome, but I have some concerns about several aspects of this study as highlighted.
Major:
1. You showed no changes in body weight between you control and metabolic syndrome groups at any checked time point, while changes in intra-abdominal fat were observed starting 3 months on the 30% sucrose diet. In the Discussion section you speculate that “The increase could be accompanied by decrease in muscle mass in the MS rats since there were modifications in body weight.” Did you checked muscle mass in those rats? Moreover, you suggest that “This may be due to a diminished consumption of solid food the MS rats, which results in a decreased protein and mineral intake in spite of higher calorie intake”, while your data presented showed a significant increase of food consumption in 5 months’ time point in rats with metabolic syndrome compared to their controls and compared to previous time points.
2. As 5 months’ time point is very well characterized in the literature data, why did you choose it as your final time point? Do you know what happens to rats at 6+ months? What age those rats die naturally? Is the metabolic syndrome main cause of death? These data would strengthen the study.
3. All presented data are lacking a baseline time point, i.e. 0 months on 30% sucrose diet. Moreover, it is not clear from the Methods section on what age rats started to receive water with sucrose.
4. It is not clear, if bar graphs presented on Figures 6 and 7 reflect the fold change to GAPDH. If so, all graphs from the control groups should start at 1.
5. Your data on LPL and HSL activity and HSL expression showed presence of difference between control rats and rats with metabolic syndrome. However, it is no changes in time in activity and expression of LPL and HSL in rats with metabolic syndrome per se. How do you explain it? Your data seem to contradict to published ones. Additionally, HSL activity presented on Figure 5, B seems to be significant starting on month 1, which is not reported.
6. It would be good to know if HSL phosphorylation is changed in rats with metabolic syndrome, as you mentioned that literature data suggest that HSL phosphorylation is increased in state of insulin resistance.
7. Many blots need to be revised. For instance, blot on Figure 6, A for MS is not representative compared to your quantification bars. Blot on Figure 7, B is overexposed.
8. Discussion section looks more like a review paper, which should be shorten. Incorporation of data obtained in the present study should be discussed instead of description of what is known from the literature. Moreover, some mistakes in description of study data in the Discussion session are presented. You mention that “…but significantly increases the serum insulin concentration are present, from the second month on.”, while data in Table 2 shows significant changes starting month 2. Another discrepancy is “…where leptin and TG significantly increased from the third month on”, while you showed significant changes on leptin starting month 2 and on TG starting on month 1.
9. When you say “Here we demonstrated that our model is characterized by a progressive increase in serum NEFAs throughout the months of treatment with 30% sucrose [54]”, using a reference at the end of this sentence makes me doubt that this is original data of the presented study.
Minor:
1. In the Introduction section you stated that “In our laboratory, we have developed a MS model in Wistar rats that is induced by chronic administration of 30% sucrose in the drinking water…”. Please add a reference to the original publication.
2. Describing adipocytes isolation, you mentioned using a kit with a 50 mesh. Did you mean a size of a strainer pore? Please clarify.
3. Describing unit that you use to express data on LPL and HSL activity, in the Methods section you mentioned nmol of free glycerol per g of tissue. However, you figure reflects it as nmol of glycerol per mg of protein. Please explain.
4. Table 2: please write variables in English.
5. Please mention Figure 1, 2 and 3 as three different figures with figure legend for each figure or make it as one figure with sub-figures A, B and C. Additionally, Figures 1-3 and 4 may be combined as one big figure for better presentation.
6. Figure 6: please correct the figure legend trying to avoid using “(for example, A = 1 month and so on)”.
To summarize, this reviewer is not convinced of the relevance of the conducted study. Observing changes in different characteristics of the metabolic syndrome in time do not contribute a lot to what has been already known at 5 months on a diet, as it is no big differences were noticed in earlier time points compared to months 5.
Author Response
Review 1
Comments and Suggestions for Authors
In their manuscript entitled “Progressive alteration of enzymes involved in lipolysis and lipogenesis that lead to adipocyte hypertrophy in intra-abdominal fat in metabolic syndrome rats” by Perez-Torres et al., the authors investigated how activity and expression of enzymes involved in the lipolysis and lipogenesis are changed in time, using a metabolic syndrome rat model on 30% sucrose in drinking water diet. The present study is focused on progressive changes in some characteristics of the metabolic syndrome, making it an observational study with no proposed mechanism involved. Overall, I think this may contribute to our knowledge about this particular metabolic syndrome, but I have some concerns about several aspects of this study as highlighted.
Major:
1. Question: You showed no changes in body weight between you control and metabolic syndrome groups at any checked time point, while changes in intra-abdominal fat were observed starting 3 months on the 30% sucrose diet. In the Discussion section you speculate that “The increase could be accompanied by decrease in muscle mass in the MS rats since there were modifications in body weight.” Did you checked muscle mass in those rats? Moreover, you suggest that “This may be due to a diminished consumption of solid food the MS rats, which results in a decreased protein and mineral intake in spite of higher calorie intake”, while your data presented showed a significant increase of food consumption in 5 months’ time point in rats with metabolic syndrome compared to their controls and compared to previous time points.
Answer:The reviewer is correct in his/her appreciation that there were no significant changes in body weight; however there was a clear tendency to an increase in weight in the MS5 group. This is now stated in the document. The tendency of weight to increase at this stage suggests the explanations proposed and we are planning to do future experiments to prove this point.
2. Question: As 5 months’ time point is very well characterized in the literature data, why did you choose it as your final time point? Do you know what happens to rats at 6+ months? What age those rats die naturally? Is the metabolic syndrome main cause of death? These data would strengthen the study.
Answer: The rats explored at 5 months of treatment were already 6 months old since the treatment began after weaning which takes place on postnatal day 21. The rats at the beginning of treatment were already weighing 100 g. Our group has previously reported the characteristics of the MS rats during aging (12 and 18 month old rats). Very few healthy rats (2 out of 10) may reach the age of approximately 24 months which would correspond to 110 years in humans but most die earlier. MS rats have s shorter lifespan than healthy ones. We had difficulty in completing our 18 month old rat group for the previously published work. (Rubio ME, Baños G, Díaz E, Guarner V. Effect of age on insulin-induced endothelin release and vasoreactivity in hypertriglyceridemic and hypertensive rats. Exp Gerontol. 2006 Mar;41(3):282-8).
3. Question: All presented data are lacking a baseline time point, i.e. 0 months on 30% sucrose diet. Moreover, it is not clear from the Methods section on what age rats started to receive water with sucrose.
Answer: The baseline time point would correspond to the weaning moment in which the rats were divided into the C and MS groups. The body and biochemical characteristics of weaning rats have been previously published by our groups when comparing control and rats receiving sucrose for a critical window during weaning. (Villegas-Romero M, Castrejón-Téllez V, Pérez-Torres I, Rubio-Ruiz ME, Carreón-Torres E, Díaz-Díaz E, Del Valle-Mondragón L, Guarner-Lans V. Short-Term Exposure to High Sucrose Levels near Weaning Has a Similar Long-Lasting Effect on Hypertension as a Long-Term Exposure in Rats. Nutrients. 2018 Jun 6;10(6). pii: E728. doi: 10.3390/nu10060728).
4. Question: It is not clear, if bar graphs presented on Figures 6 and 7 reflect the fold change to GAPDH. If so, all graphs from the control groups should start at 1.
Answer: No, graphs do not reflect the fold change to GAPDH but the ratio of density of the stain of the protein selected/density of the stain of GAPDH.
5. Question: Your data on LPL and HSL activity and HSL expression showed presence of difference between control rats and rats with metabolic syndrome. However, it is no changes in time in activity and expression of LPL and HSL in rats with metabolic syndrome per se. How do you explain it? Your data seem to contradict to published ones. Additionally, HSL activity presented on Figure 5, B seems to be significant starting on month 1, which is not reported.
Answer: The changes in the expression and activity of enzymes are not always simultaneous. Sometimes increases in expression precede changes in activity since the expressed protein is inactive. This was the case in our experiments where expression was increased but there were no changes in NEFAs that reflected a change in the activity of the enzyme. It was not until the activity of the enzyme was increased that we found a significant change in NEFAs. Therefore, the changes were related more to activity than to expression. This is stated in the “Lipolysis and HSL section” of the discussion.
6. Question: It would be good to know if HSL phosphorylation is changed in rats with metabolic syndrome, as you mentioned that literature data suggest that HSL phosphorylation is increased in state of insulin resistance.
Answer: The reviewer is correct; unfortunately we did not have the means to measure phosphorylated HSL. However, the activity reflects the effects of the phosphorylated enzyme.
7. Question: Many blots need to be revised. For instance, blot on Figure 6, A for MS is not representative compared to your quantification bars. Blot on Figure 7, B is overexposed.
Answer: The previous blots were substituted for more representative ones.
8. Question: Discussion section looks more like a review paper, which should be shorten. Incorporation of data obtained in the present study should be discussed instead of description of what is known from the literature. Moreover, some mistakes in description of study data in the Discussion session are presented. You mention that “…but significantly increases the serum insulin concentration are present, from the second month on.”, while data in Table 2 shows significant changes starting month 2. Another discrepancy is “…where leptin and TG significantly increased from the third month on”, while you showed significant changes on leptin starting month 2 and on TG starting on month 1. Some details of the experiments that were cited were deleted to shorten the discussion.
Answer: Errors have been corrected.
9. Question: When you say “Here we demonstrated that our model is characterized by a progressive increase in serum NEFAs throughout the months of treatment with 30% sucrose [54]”, using a reference at the end of this sentence makes me doubt that this is original data of the presented study.
Answer: The reference was deleted.
Minor:
1. Question: In the Introduction section you stated that “In our laboratory, we have developed a MS model in Wistar rats that is induced by chronic administration of 30% sucrose in the drinking water…”. Please add a reference to the original publication.
Answer: A reference was added.
2. Question: Describing adipocytes isolation, you mentioned using a kit with a 50 mesh. Did you mean a size of a strainer pore? Please clarify.
Answer: The issue was clarified indicated that we used a 50 pore mesh.
3. Question: Describing unit that you use to express data on LPL and HSL activity, in the Methods section you mentioned nmol of free glycerol per g of tissue. However, you figure reflects it as nmol of glycerol per mg of protein. Please explain.
Answer: We are sorry for the error; we have corrected it.
4. Question: Table 2: please write variables in English.
Answer: Done, It is was corrected.
5. Question: Please mention Figure 1, 2 and 3 as three different figures with figure legend for each figure or make it as one figure with sub-figures A, B and C. Additionally, Figures 1-3 and 4 may be combined as one big figure for better presentation.
Answer: This point was also mentioned by another reviewer and we have left only fig 1 A, B, C, and D. Data on figure 1B are new since they were asked by the other reviewer.
6. Question: Figure 6: please correct the figure legend trying to avoid using “(for example, A = 1 month and so on)”.
Answer: We corrected the figure legend.
Question: To summarize, this reviewer is not convinced of the relevance of the conducted study. Observing changes in different characteristics of the metabolic syndrome in time do not contribute a lot to what has been already known at 5 months on a diet, as it is no big differences were noticed in earlier time points compared to months 5.
Answer: We thank the reviewer for his/her comment; however we disagree since as stated by reviewer 2 the study of the developmental timeline of the disease could help in establishing better therapeutic measures at different stages of the disease.
Review 2
Comments and Suggestions for Authors
1. Question: The reasons for using a guillotine in animal sacrifice and the process of sampling should be described more specifically.
Answer: We used a guillotine since anesthetizing the animals for this study would interfere with the functioning of adipocytes. Pentobarbital, for example is stored in fat tissue. We deleted the term guillotine and we now only state that animals were decapitated. We have included the formulas for determining the sample size. And mention that the blood was collected from the severed neck.
2. Question: Statistical significance in figures should be indicated using symbols.
Answer: We did not use symbols since we believe the way of indicating significance used in this paper is appropriate and clear.
3. Question: Checking and correcting English sentences is required.
Answer: The paper was revised by an expert English speaking person.
4. Question: The results of the tables should be rechecked.
Answer: The results in the tables were checked.
5. Question: The results of out-of-normal range (eg, TG 3 months, AGEs 4 months, etc.) and statistical analysis may also need to be checked.
Answer: We have revised the data and the reported values are correct.
6. Question: The present manuscript type is described as a review paper, so it is necessary to directly link the results of the present study (the characteristics of the MS rat model) in the discussion part. In the MS rat model, regarding changes in muscle formation mechanism through MAPK by insulin or IGF1 needed to be discussed.
Answer: thanks you for your comments. This is an original paper and not a review. In the discussion we review papers that are related to the data obtained. We have deleted some of the descriptive data of the papers cited to shorten the discussion. The changes in muscle during the establishment of MS will be the object of a future study. However, now were added two lines in the discussion section, on changes in muscle formation mechanism through MAPK by insulin and IGF-1.
7. Question: The time course of the MS model induced by this 30% sucrose chronic consumption is required to examine the academic meaning and the application field in the development of the therapeutic drug based on the comparison with the human disease and other similar models.
Answer: Thank you for your comment.
Review 3
Comments and Suggestions for Authors
Pérez-Torres et al describe some molecular changes occurring in adipocytes from abdominal adipose tissue during metabolic syndrome insurgence.
The paper is well written, methods appear to be appropriate, and so statistics; the conclusions are supported by the achieved results.
Question: However, I think that the Authors should consider also if the total adipocytes' number (i.e. cells' number normalized per intra-abdominal fat weight) changes in course of metabolic syndrome, not only the adipocytes diameter or the adipocytes' number per field. I suggest to add this information in the text ("Intra-abdominal adipocytes" paragraph) and in the "Figure 1" by adding an histogram as "Panel D".
Answer, Thank you for your comment, now we calculated and included data on total adipocyte number/g fat tissue.
Question: In the first sentence of the "Intra-abdominal adipocytes" paragraph the Authors should specify in what therm the intra-abdominal fat of the first and second month showed no significant differences: in weight? in cells' diameter?
Answer: This was corrected in the manuscript.
Question: The Authors stated that the animal model that they used has no alterations in glucose metabolism, since no significant differences in the glucose concentrations were find in any of the experimental groups (pag 5, first paragraph and pag 14, first paragraph). I desagree with this statement, because alterations in HOMA index, insulin and AGEs concentrations (three parameters that are direct and/or indirect expression of glucose metabolism) were reported in the present study. Please, correct and clarify this point, and discuss it properly.
Answer, Thank you for your comment, we have deleted the sentences in which there were no alterations in glucose metabolism. Our model is a MS model and not a model of diabetes and therefore glucose metabolism alterations are not as significant. However, there are still changes in glucose and TG that are definitely a reflection of changes in glucose metabolism.
Question: Figures, 1, 2 and 3 should be joined and presented as separate panels (A, B, C) of the same figure 1, where I suggest to add also the information about the adipocytes' number normalized per intra-abdominal fat weight. Figures' number should be modified consequently. Adipocytes' diameter should be reported as μm, not in pixels.
Answer: The figure was modified as suggested and changes in adipocyte number for gram of tissue were included.
Question: Concerning the Figure 4, no histogram is presented, differently from what stated in the legend. Moreover, the sentence "Histological results were associated with the amount of intra-abdominal fat, adipocyte cell number per field, and adipocyte diameter" is not clear to me: what do the Authors mean with "associated"? In what terms the histological results are associated with intra-abdominal fat? I suggest to provide this figure as a supplemental one, with the description now included in the "results" section (page 5, last paragraph). Infact, the last paragraph of page 5 (from "Figure 4 shows..." to "...clearly show adipocyte hypertrophy") should be removed from the main text since all the described data are clearly reported in the previous figures.
Answer: The figure legend was modified to avoid the issues raised by the reviewer.
Question: In the first paragraph of page 8 a significant increase in LPL activity is reported also in the first month of observation, but a p=0.05 does not indicate a statistically significant difference. Please, correct.
Answer: The cutting point for a statistically significant difference is p≤0.05. Therefore a p=0.05 is statistically significant.
Question: No significant difference between HSL expression of the MS vs. C rats during the first month is reported in the second paragraph of page 8, but in the Figure 7A a p=0.04 is reported. Please, clarify and correct.
Answer: The observation of the reviewer is correct and the issue was modified.
Question: Some confusion with figures' number is present in page 8. Figure 6 is Fig 5B, fig 7 is figure 6, and figure 8 is figure7 instead. But the numbers should be entirely revised if the suggestion to join fig 1, 2, 3 and to remove figure 4 is accepted.
Answer: Figure numbering was corrected.
Question: Some explanations are required regarding some sentences in the discussion (page 15):
"However, the LPL expression then remains unchanged and therefore, the expression of this enzyme is inversely related to its activity"
Answer: This was modified.
"Although activity and expression of HSL may be directly associated, the lipolytic capacity of this enzyme is usually related to its activity"
Answer: This is now clarified.
I think that these sentences should be rephrased because the sense of them appear to be not clear to me in the present form.
Answer: Done.
Question: In page 14, the Authors suggest that rats probably develop sarcopenia, but this hypothesis deserves further investigations (i.e. imaging) according to me.
Answer: We agree with the reviewer and are currently working on another paper addressing the development of sarcopenia in this model.
Question: Please, abbreviate total cholesteros as TC, not CT. Correct through the whole paper.
Answer: Done.
Question: In M&M, "Serum Biochemical Variables" section, the Authors stated that Commercially obtained ELISA kits were used for the determination of some serum biochemical variables: which ones? Please specify. Please correct "glucose mM/L" with "glucose mmol/L".
Answer: The kits used are specified in the following sentences to the one the reviewer is citing. We changed the units in which glucose was expressed as suggested.
Question: In Table 2 I think that there was an error in the value of TG, 3 months - C Group: 2.6+/- 2.0. Please, correct.
Answer: The reviewer is correct and we have substituted the value for the correct one.
Question: I think that the title of the paper should be more concise and clear; please, provide a new one.
Answer: The title was modified for a more concise one.
Thanks you for your comments.

Reviewer 2 Report
1. The reasons for using a guillotine in animal sacrifice and the process of sampling should be described more specifically.
2. Statistical significance in figures should be indicated using symbols.
3. Checking and correcting English sentences is required.
4. The results of the tables should be rechecked.
5. The results of out-of-normal range (eg, TG 3 months, AGEs 4 months, etc.) and statistical analysis may also need to be checked.
6. The present manuscript type is described as a review paper, so it is necessary to directly link the results of the present study (the characteristics of the MS rat model) in the discussion part. In the MS rat model, regarding changes in muscle formation mechanism through MAPK by insulin or IGF1 needed to be discussed.
7. The time course of the MS model induced by this 30% sucrose chronic consumption is required to examine the academic meaning and the application field in the development of the therapeutic drug based on the comparison with the human disease and other similar models.
Author Response

(The authors gave the same response as above.)

Reviewer 3 Report
Pérez-Torres et al describe some molecular changes occurring in adipocytes from abdominal adipose tissue during metabolic syndrome insurgence.
The paper is well written, methods appear to be appropriate, and so statistics; the conclusions are supported by the achieved results.
- However, I think that the Authors should consider also if the total adipocytes' number (i.e. cells' number normalized per intra-abdominal fat weight) changes in course of metabolic syndrome, not only the adipocytes diameter or the adipocytes' number per field. I suggest to add this information in the text ("Intra-abdominal adipocytes" paragraph) and in the "Figure 1" by adding an histogram as "Panel D".
- In the first sentence of the "Intra-abdominal adipocytes" paragraph the Authors should specify in what therm the intra-abdominal fat of the first and second month showed no significant differences: in weight? in cells' diameter?
- The Authors stated that the animal model that they used has no alterations in glucose metabolism, since no significant differences in the glucose concentrations were find in any of the experimental groups (pag 5, first paragraph and pag 14, first paragraph). I desagree with this statement, because alterations in HOMA index, insulin and AGEs concentrations (three parameters that are direct and/or indirect expression of glucose metabolism) were reported in the present study. Please, correct and clarify this point, and discuss it properly.
- Figures, 1, 2 and 3 should be joined and presented as separate panels (A, B, C) of the same figure 1, where I suggest to add also the information about the adipocytes' number normalized per intra-abdominal fat weight. Figures' number should be modified consequently. Adipocytes' diameter should be reported as μm, not in pixels.
- Concerning the Figure 4, no histogram is presented, differently from what stated in the legend. Moreover, the sentence "Histological results were associated with the amount of intra-abdominal fat, adipocyte cell number per field, and adipocyte diameter" is not clear to me: what do the Authors mean with "associated"? In what terms the histological results are associated with intra-abdominal fat? I suggest to provide this figure as a supplemental one, with the description now included in the "results" section (page 5, last paragraph). Infact, the last paragraph of page 5 (from "Figure 4 shows..." to "...clearly show adipocyte hypertrophy") should be removed from the main text since all the described data are clearly reported in the previous figures.
- In the first paragraph of page 8 a significant increase in LPL activity is reported also in the first month of observation, but a p=0.05 does not indicate a statistically significant difference. Please, correct.
- No significant difference between HSL expression of the MS vs. C rats during the first month is reported in the second paragraph of page 8, but in the Figure 7A a p=0.04 is reported. Please, clarify and correct.
- Some confusion with figures' number is present in page 8. Figure 6 is Fig 5B, fig 7 is figure 6, and figure 8 is figure7 instead. But the numbers should be entirely revised if the suggestion to join fig 1, 2, 3 and to remove figure 4 is accepted.
- Some explanationes are required regarding some sentences in the discussion (page 15):
- "However, the LPL expression then remains unchanged and therefore, the expression of this enzyme is inversely related to its activity"
- "Although activity and expression of HSL may be directly associated, the lipolytic capacity of this enzyme is usually related to its activity"
I think that these sentences should be rephrased because the sense of them appear to be not clear to me in the present form.
- In page 14, the Authors suggest that rats probably develop sarcopenia, but this hypothesis deserves further investigations (i.e. imaging) according to me.
- Please, abbreviate total cholesteros as TC, not CT. Correct through the whole paper.
- In M&M, "Serum Biochemical Variables" section, the Authors stated that Commercially obtained ELISA kits were used for the determination of some serum biochemical variables: which ones? Please specify. Please correct "glucose mM/L" with "glucose mmol/L".
- In Table 2 I think that there was an error in the value of TG, 3 months - C Group: 2.6+/- 2.0. Please, correct.
- I think that the title of the paper should be more concise and clear; please, provide a new one
Author Response

(The authors gave the same response as above.)

Round 2
Reviewer 3 Report
The Authors answered to all the questions raised by the Reviewer.
Only a clarification is still due: a null hypothesis is rejected if the p-value is lower than, and not equal to, the chosen significance level of the test. I still suggest that is mandatory to correct this item in the results relative to LPL activity
Author Response
Thanks you for your comments.
Question 1:
English language and style are fine/minor spell check required
Answer
Done, English language was checked and corrected.
Question 2:
Only a clarification is still due: a null hypothesis is rejected if the p-value is lower than, and not equal to, the chosen significance level of the test. I still suggest that is mandatory to correct this item in the results relative to LPL activity
Answer:
The observation for the referee it is correct, now our modified the text in result section and delete the significant change in the figure 3, in where was show LPL activity.
